METHODS AND RESOURCES

# Estimating the transfer rates of bacterial plasmids with an adapted Luria–Delbrück fluctuation analysis

Olivia Kosterlitz[1,2]*, Adamaris Muñiz Tirado[1], Claire Wate[1], Clint Elg[2,3], Ivana Bozic[4], Eva M. Top[2,3], Benjamin Kerr[1,2]*

1 Biology Department, University of Washington, Seattle, Washington, United States of America, 2 BEACON Center for the Study of Evolution in Action, East Lansing, Michigan, United States of America, 3 Department of Biological Sciences and Institute for Interdisciplinary Data Sciences, University of Idaho, Moscow, Idaho, United States of America, 4 Department of Applied Mathematics, University of Washington, Seattle, Washington, United States of America

* livkost@uw.edu (OK); kerrb@uw.edu (BK)

## Abstract

To increase our basic understanding of the ecology and evolution of conjugative plasmids, we need reliable estimates of their rate of transfer between bacterial cells. Current assays to measure transfer rate are based on deterministic modeling frameworks. However, some cell numbers in these assays can be very small, making estimates that rely on these numbers prone to noise. Here, we take a different approach to estimate plasmid transfer rate, which explicitly embraces this noise. Inspired by the classic fluctuation analysis of Luria and Delbrück, our method is grounded in a stochastic modeling framework. In addition to capturing the random nature of plasmid conjugation, our new methodology, the Luria–Delbrück method ("LDM"), can be used on a diverse set of bacterial systems, including cases for which current approaches are inaccurate. A notable example involves plasmid transfer between different strains or species where the rate that one type of cell donates the plasmid is not equal to the rate at which the other cell type donates. Asymmetry in these rates has the potential to bias or constrain current transfer estimates, thereby limiting our capabilities for estimating transfer in microbial communities. In contrast, the LDM overcomes obstacles of traditional methods by avoiding restrictive assumptions about growth and transfer rates for each population within the assay. Using stochastic simulations and experiments, we show that the LDM has high accuracy and precision for estimation of transfer rates compared to the most widely used methods, which can produce estimates that differ from the LDM estimate by orders of magnitude.

## Introduction

A fundamental rule of heredity involves the passage of genes from parents to their offspring. Bacteria violate this rule of strict vertical inheritance by shuttling DNA between cells through horizontal gene transfer [1,2]. Often the genetic elements being shuttled are plasmids,

**Data Availability Statement:** All generated data and custom software are deposited in a GitHub repository (https://github.com/livkosterlitz/LDM)

and archived on Zenodo (https://doi.org/10.5281/zenodo.6677158).

**Funding:** E.M.T. and B.K received support for this work from National Institute of Allergy and Infectious Diseases Extramural Activities grant no. R01 AI084918 from the National Institutes of Health. B.K received support for this work from Division of Environmental Biology grant no. 2142718 from the National Science Foundation. O.K. was supported by the National Science Foundation Graduate Research Fellowship grant no. DGE-1762114. C.E. was supported by the National Science Foundation Graduate Research Fellowship grant no. DGE-2019265372. The funders had no role in study design, data collection and analysis, decision to publish, or preparation of the manuscript.

**Competing interests:** The authors have declared that no competing interests exist.

extrachromosomal DNA molecules that can encode the machinery for their transfer [3]. This plasmid transfer process is termed conjugation, in which a plasmid copy is moved from one cell to another upon direct contact. Additionally, plasmids replicate independently inside their host cell to produce multiple copies, which segregate into both offspring upon cell division. Therefore, conjugative plasmids are governed by 2 modes of inheritance: vertical and horizontal.

This horizontal mode of inheritance makes it possible for nonrelated cells to exchange genetic material, which includes members of different species [4]. In fact, conjugation can occur across vast phylogenetic distances, such that the expansive gene repertoire in the "accessory" genome encoded on conjugative plasmids is shared among many microbial species [5]. This ubiquitous genetic exchange reinforces the central role of conjugation in shaping the ecology and evolution of microbial communities [1,3,6]. Notably, conjugation is a common mechanism facilitating the spread of antimicrobial resistance genes among bacteria and the emergence of multidrug resistance in clinical pathogens [7–9]. To understand how genes, including those of clinical relevance, move within complex bacterial communities, an accurate and precise measure of the rate of conjugation is of the utmost importance.

The basic approach to measure conjugation involves mixing plasmid-containing bacteria, called "donors," with plasmid-free bacteria, called "recipients." As the coculture incubates, recipients acquire the plasmid from the donor through conjugation, and these transformed recipients are called "transconjugants." Over the course of this "mating assay," the densities of donors, recipients, and transconjugants are tracked over time ($D_t$, $R_t$, and $T_t$, respectively) as the processes of population growth and plasmid transfer occur. To understand how such information is used to calculate the rate of conjugation, we consider an altered version of the foundational model of Levin and colleagues [10]. In this framework, populations grow exponentially, and recipients become transconjugants via conjugation when they interact with plasmid-bearing cells (i.e., donors or transconjugants). The densities of the populations are described by the following differential equations (the $t$ subscript is dropped from the variables for notational convenience):

$$\frac{dD}{dt} = \psi_D D, \tag{1}$$

$$\frac{dR}{dt} = \psi_R R - \gamma_D D R - \gamma_T T R, \tag{2}$$

$$\frac{dT}{dt} = \psi_T T + \gamma_D D R + \gamma_T T R. \tag{3}$$

In Eqs 1–3, donors, recipients, and transconjugants divide at a per capita rate of $\psi_D$, $\psi_R$, and $\psi_T$, respectively. The parameters $\gamma_D$ and $\gamma_T$ measure the rate at which a recipient acquires a plasmid per unit density of the donor and transconjugant, respectively. Thus, the $\psi$ parameters are population growth rates, and the $\gamma$ parameters are conjugation rates (see Fig 1A). Assuming all the growth rates are equal ($\psi_D = \psi_R = \psi_T = \psi$) and conjugation rates are equal ($\gamma_D = \gamma_T = \gamma$), Simonsen and colleagues [11] provided an elegant solution to Eqs 1–3 to produce the following estimate for the conjugation rate from donors to recipients (hereafter termed the "donor conjugation rate"):

$$\gamma_D = \psi \ln\left(1 + \frac{T_{\tilde{t}}}{R_{\tilde{t}}} \frac{N_{\tilde{t}}}{D_{\tilde{t}}}\right) \frac{1}{(N_{\tilde{t}} - N_0)}. \tag{4}$$

For a mating assay incubated for a fixed period (hereafter $\tilde{t}$), the initial and final density of all

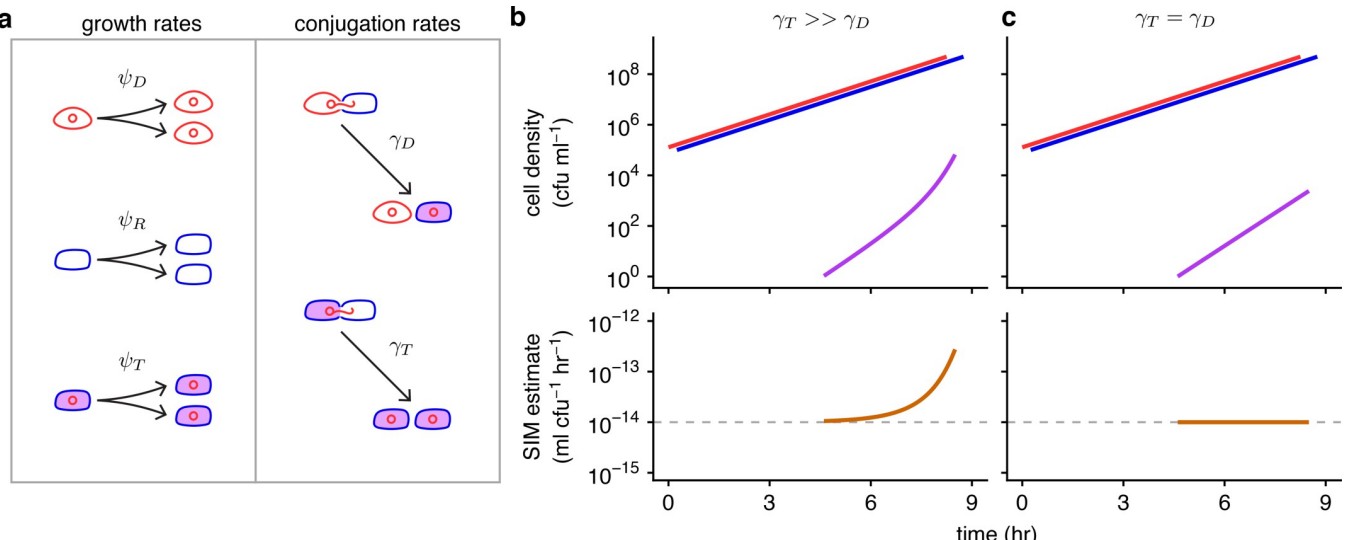

**Fig 1. Basic model parameters and the effects of unequal conjugation rates on the SIM estimate.** (a) In this schematic, the conjugative plasmid is a red circle, a donor is a red cell containing the plasmid, a recipient is a blue cell, and a transconjugant is indicated with a purple interior (a blue cell containing a red plasmid). The $\psi_D$, $\psi_R$, and $\psi_T$ parameters are donor, recipient, and transconjugant growth rates, respectively, illustrated by 1 cell dividing into 2. The $\gamma_D$ and $\gamma_T$ parameters are donor and transconjugant conjugation rates, respectively, shown by conjugation events transforming recipients into transconjugants. (b) When the transconjugant conjugation rate ($\gamma_T$) is higher than the donor conjugation rate ($\gamma_D$), transconjugants exhibit superexponential increase (purple curve) while donors and recipients increase exponentially (red and blue lines). The SIM estimate (orange line) increases over time, deviating from the actual donor conjugation rate (gray dashed line). (c) In contrast, when the conjugation rates are equal ($\gamma_T = \gamma_D$), the transconjugant increase is muted relative to part b (purple line). The SIM assumptions are met, and the estimate is constant and accurate over time (orange line). Eqs 1–3 were used to produce the top graphs, with $D_0 = R_0 = 10^5$, $T_0 = 0$, $\psi_D = \psi_R = \psi_T = 1$, $\gamma_D = 10^{-14}$, and either $\gamma_T = 10^{-8}$ (in part b) or $\gamma_T = 10^{-14}$ (in part c). The donor and recipient trajectories overlapped but were staggered for visibility. Eq 4 was used to produce the bottom graphs. The code needed to generate this figure can be found at https://github.com/livkosterlitz/LDM or https://doi.org/10.5281/zenodo.6677158.

bacteria ($N_0$ and $N_{\bar{i}}$, respectively), the final density of each cell population ($D_{\bar{i}}$, $R_{\bar{i}}$, and $T_{\bar{i}}$), and the population growth rate ($\psi$) are sufficient for an estimate of the conjugation rate.

The Achilles heel of this estimate, as with others, is found in violations of its assumptions. For example, we label Eq 4 as the "Simonsen and colleagues Identicality Method" estimate (SIM) for the donor conjugation rate because the underlying model assumes all strains are identical with regards to growth rates and conjugation rates. However, in natural microbial communities, this identicality assumption is misplaced, especially when the donors and recipients belong to different species. For example, suppose that the rate of plasmid transfer within a species (i.e., from transconjugants to recipients, which we abbreviate as the "transconjugant conjugation rate") is much higher than between species (i.e., from donors to recipients), i.e., $\gamma_T \gg \gamma_D$ (Fig 1B). This elevated within-species conjugation rate ($\gamma_T$) will increase the number of transconjugants and consequently inflate the SIM estimate for the cross-species conjugation rate ($\gamma_D$) compared to a case where the conjugation rates are equal ($\gamma_T = \gamma_D$, Fig 1C). This Achilles heel is not specific to cross-species scenarios and can occur when estimating conjugation between any cells, including strains of the same species. One approach to minimize the resulting bias is to shorten the incubation time for the assay [12], as estimate bias tends to increase over time (e.g., Fig 1B). However, new problems can arise when using this approach, such as the transconjugant numbers being exceedingly low and thus difficult to accurately assess [13]. Another approach was introduced by Huisman and colleagues [14], which squarely addressed the SIM identicality assumptions by developing a method to estimate donor conjugation rate when growth and transfer rates differ, thereby enlarging the set of systems amenable to estimation (see Section 1 in S1 File for full description of this and other approaches).

Nonetheless, this new method can have difficulty with situations in which the donor conjugation rate ($\gamma_D$) is substantially lower than the transconjugant rate ($\gamma_T$), such as the example illustrated in Fig 1. Such differences have been reported in multispecies systems [15] and recently several studies have recognized the importance of evaluating the biology of plasmids in microbial communities [7,16–18]. Therefore, a method that provides an accurate estimate despite substantial inequalities in rate parameters is desirable.

Here, we derive a novel estimate for conjugation rate by explicitly tracking transconjugant dynamics as a stochastic process (i.e., a continuous time branching process). This represents a notable deviation from previous approaches that are built upon deterministic frameworks. The random nature of conjugation can lead to substantial variation in the number of transconjugants at the end of a mating assay ($T_i$) as this population will often be small. Prior deterministic frameworks rely on this number (e.g., Eq 4), such that transconjugant variation adds problematic noise to the estimate. In contrast, our stochastic approach leverages this noise to produce an estimate (akin to the way Luria and Delbrück estimated mutation rate [19]). In addition, our method allows for unrestricted heterogeneity in growth rates and conjugation rates. Thus, our method fills a gap in the methodological toolkit by allowing unbiased estimation of conjugation rates in a wide variety of strains and species. We used stochastic simulations to validate our estimate and compare its accuracy and precision to other estimates. We developed a protocol for the laboratory by using microtiter plates to rapidly screen many donor-recipient cocultures for the existence of transconjugants. In addition to its experimental tractability, our protocol circumvents problems that arise in the laboratory that can bias other approaches. Finally, we implemented our method in the laboratory and compared our estimate to the SIM estimate using a *Klebsiella pneumoniae* to *Escherichia coli* cross-species case study with an IncF conjugative plasmid.

## Results

### A new conjugation rate estimate inspired by the Luria–Delbrück approach

Previous methods to estimate the rate of conjugation have treated the rise of transconjugants as a deterministic process (i.e., nonrandom). However, conjugation is inherently a stochastic (i.e., random) process [20]. Given that conjugation transforms the genetic state of a cell, we can form an analogy with mutation, which is also a stochastic process that transforms the genetic state of a cell. While mutation transforms a wild-type cell to a mutant, conjugation transforms a recipient cell to a transconjugant.

This analogy inspired us to revisit the way Luria and Delbrück handled the mutational process in their classic paper on the nature of bacterial mutation [19], outlined in Fig 2A–2D. For this process, assume that the number of wild-type cells, $N_t$, is expanding exponentially. Let the rate of mutant formation be given by $\mu$. In Fig 2A, we see that the number of mutants in a growing population increases due to mutation events (highlighted purple cells) and due to faithful reproduction by mutants (nonhighlighted purple cells). The rate at which mutants are generated (highlighted purple cells) is $\mu N_t$, which grows as the number of wild-type cells increase (Fig 2B). However, the rate of transformation per wild-type cell is the mutation rate $\mu$, which is constant (Fig 2C). Since mutations are random, parallel cultures will vary in the number of mutants depending on if and when mutation events occur. As seen in Fig 2D, for sufficiently small wild-type populations growing over sufficiently small periods, some replicate populations will not contain any mutant cell (gray shading) while other populations exhibit mutants (purple shading). Indeed, the cross-replicate fluctuation in the number of mutants was a critical component of the Luria–Delbrück experiment.

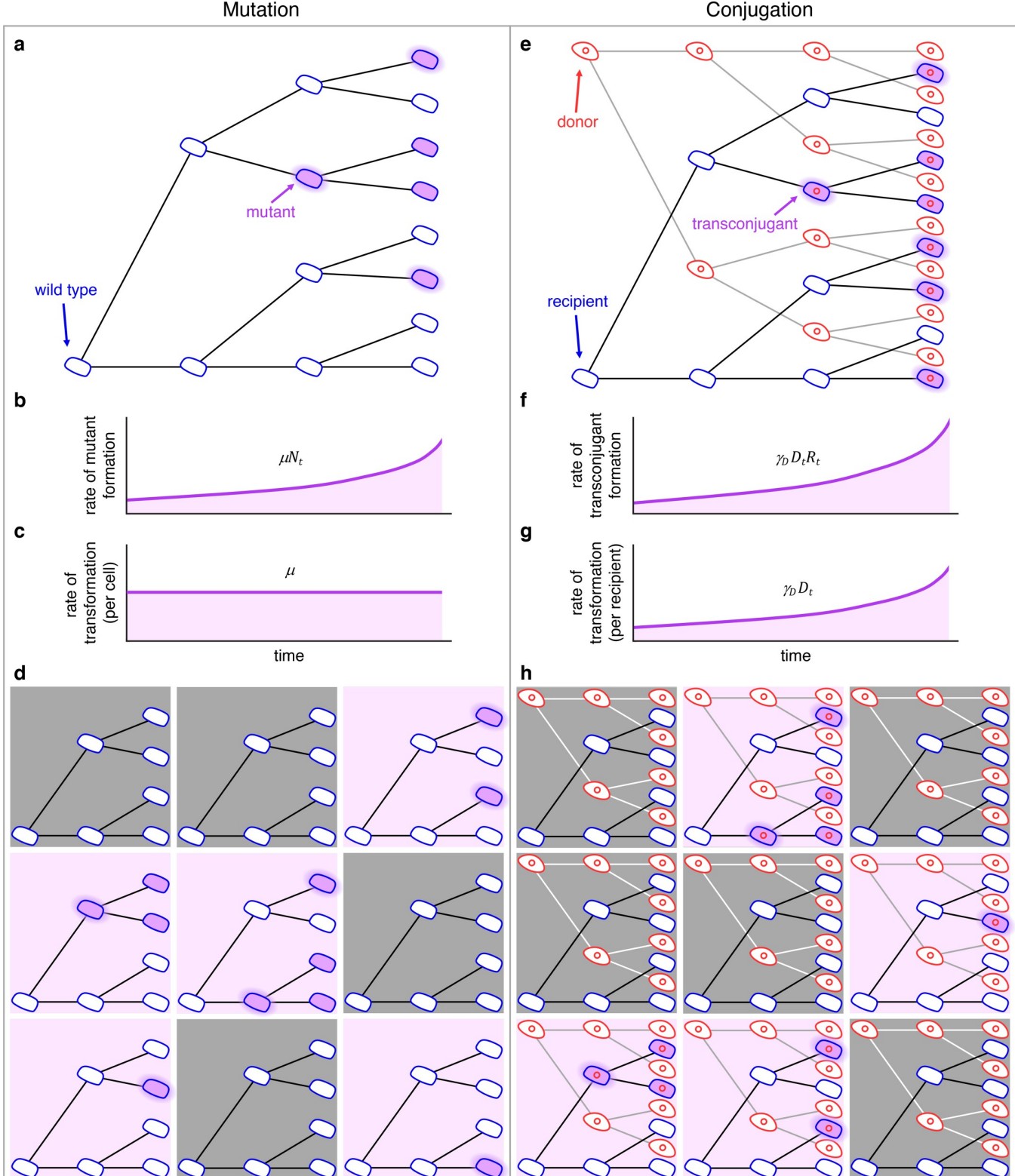

**Fig 2. Schematic comparing the process of mutation (a–d) to the process of conjugation (e–h).** (a) In a growing population of wild-type cells, mutants arise (highlighted purple cells) and reproduce (nonhighlighted purple cells). (b) The rate at which mutants are generated grows as the number of wild-type cells increases (i.e., $\mu N_t$). (c) The rate of transformation per wild-type cell is the mutation rate $\mu$. (d) Wild-type cells growing in 9 separate populations where mutants arise in a portion of the populations (those with purple backgrounds) at different cell divisions. (e) In a growing population of donors and recipients,

transconjugants arise (highlighted purple cells) and reproduce (nonhighlighted purple cells). (f) The rate at which transconjugants are generated grows as the numbers of donors and recipients increase (i.e., $\gamma_D D_t R_t$). (g) The rate of transformation per recipient cell grows as the number of donors increases (i.e., $\gamma_D D_t$), where $\gamma_D$ is the constant conjugation rate parameter. (h) Donor and recipient cells growing in 9 separate populations where transconjugants arise in a portion of the populations (purple backgrounds) at different points in time. For all panels, this is a conceptual figure, and the rates are inflated for illustration purposes.

To apply this strategy to estimate the conjugation rate, we can similarly think about an exponentially growing population of recipients (Fig 2E). But now there is another important cell population present (the donors). The transformation of a recipient is simply the generation of a transconjugant (highlighted purple cells) via conjugation with a donor. If we ignore conjugation from transconjugants for the moment, the rate at which transconjugants are generated is $\gamma_D D_t R_t$ (Fig 2F). In contrast to the mutation rate, the rate of transformation per cell is not a constant. Rather, this transformation rate per recipient is $\gamma_D D_t$, which grows with the donor population (Fig 2G). It is as if we are tracking a mutation process where the mutation rate is exponentially increasing. Yet the rate of transformation per recipient and donor is constant, which is the donor conjugation rate $\gamma_D$. As with mutation, conjugation is random, which results in a distribution in the number of transconjugants among parallel cultures depending on the time points at which transconjugants arise. As seen in Fig 2H, under certain conditions, some replicate populations will not contain any transconjugant cell (gray shading), while other populations will exhibit transconjugants (purple shading).

Using this analogy, here we describe a new approach for estimating conjugation rate that embraces conjugation as a stochastic process [20]. Let the density of donors, recipients, and transconjugants in a well-mixed culture at time $t$ be given by the variables $D_t$, $R_t$, and $T_t$. In all that follows, we will assume that the culture is inoculated with donors and recipients, while transconjugants are initially absent (i.e., $D_0 > 0$, $R_0 > 0$, and $T_0 = 0$). The donor and recipient populations grow according to the following standard exponential growth equations

$$D_t = D_0 e^{\psi_D t}, \tag{5}$$

$$R_t = R_0 e^{\psi_R t}, \tag{6}$$

where $\psi_D$ and $\psi_R$ are the growth rates for donor and recipient cells, respectively. With Eqs 5 and 6, we are making a few assumptions, which also occur in some of the previous methods (Table C in S1 File). First, we assume the loss of recipient cells to transformation into transconjugants can be ignored. This assumption is acceptable because, for what follows, the rate of generation of transconjugants per recipient cell (as in Fig 2G, $\gamma_D D_t$) is very small relative to the per capita recipient growth rate ($\psi_R$). Second, we assume that donors and recipients exhibit deterministic exponential growth. If the initial numbers of donors and recipients are not too small (i.e., $D_0 \gg 0$ and $R_0 \gg 0$) and per capita growth remains constant over the period of interest, then this assumption is reasonable. We note that this assumption does not deny that cell division of donors and recipients are also stochastic processes, but given the large numbers of these cells, a deterministic approximation is appropriate.

On the other hand, the number of transconjugants over the period of interest can be quite small (starting from zero), motivating an explicit stochastic treatment [21]. The population growth of transconjugants is modeled using a continuous time stochastic process. The number of transconjugants, $T_t$, is a random variable taking on nonnegative integer values. In this section, we will assume the culture volume is 1 ml, and thus, the number of transconjugants is equivalent to the density of transconjugants (per ml). For a very small interval of time, $\Delta t$, the current number of transconjugants will either increase by one or remain constant. The

probabilities of each possibility are given as follows:

$$\Pr\{T_{t+\Delta t} = T_t + 1\} = \gamma_D D_t R_t \Delta t + \gamma_T T_t R_t \Delta t + \psi_T T_t \Delta t, \tag{7}$$

$$\Pr\{T_{t+\Delta t} = T_t\} = 1 - (\gamma_D D_t R_t + \gamma_T T_t R_t + \psi_T T_t)\Delta t. \tag{8}$$

The 3 terms on the right-hand side of Eq 7 illustrate the processes enabling the transconjugant population to increase. The first term gives the probability that a donor transforms a recipient into a transconjugant via conjugation. The second term gives the probability that a transconjugant transforms a recipient via conjugation. The third term measures the probability that a transconjugant cell divides. Eq 8 is simply the probability that none of these 3 processes occur.

Given the standard setup of a mating assay, we focus on a situation where there are initially no transconjugants. Therefore, the only process that can produce the first transconjugant is conjugation of the plasmid from a donor to a recipient. Using Eq 8 with $T_t = 0$, we have

$$\Pr\{T_{t+\Delta t} = 0|T_t = 0\} = 1 - \gamma_D D_t R_t \Delta t. \tag{9}$$

We let the probability that we have zero transconjugants at time $t$ be denoted by $p_0(t)$ (i.e., $p_0(t) = \Pr\{T_t = 0\}$). In Section 2 in S1 File, we derive the following expression for $p_0(t)$ at time $t = \tilde{t}$:

$$p_0(\tilde{t}) = \exp\left\{\frac{-\gamma_D D_0 R_0}{\psi_D + \psi_R}\left(e^{(\psi_D + \psi_R)\tilde{t}} - 1\right)\right\}. \tag{10}$$

Solving Eq 10 for $\gamma_D$ yields a new measure for the donor conjugation rate:

$$\gamma_D = -\ln p_0(\tilde{t})\left(\frac{\psi_D + \psi_R}{D_0 R_0 (e^{(\psi_D + \psi_R)\tilde{t}} - 1)}\right). \tag{11}$$

This expression is similar in form to the mutation rate derived by Luria and Delbrück in their classic paper on the nature of bacterial mutation [19], which is not a coincidence.

In Section 3 in S1 File, we rederive the Luria–Delbrück result, which can be expressed as

$$\mu = -\ln p_0(\tilde{t})\left(\frac{\psi_N}{N_0 (e^{\psi_N \tilde{t}} - 1)}\right). \tag{12}$$

In the mutational process modeled by Luria and Delbrück, $N_0$ is the initial wild-type population size, which grows exponentially at rate $\psi_N$. For Luria and Delbrück, $p_0(\tilde{t})$ refers to the probability of zero mutants at time $\tilde{t}$ (as in a gray-shaded tree in Fig 2D), whereas $p_0(\tilde{t})$ in the conjugation estimate refers to the probability of zero transconjugants (as in a gray-shaded tree in Fig 2H). Comparing Eq 12 to Eq 11, conjugation can be thought of as a mutation process with initial wild-type population size $D_0 R_0$ that grows at rate $\psi_D + \psi_R$. The structural similarity of the estimates is grounded in a structural similarity of the underlying models; indeed, some of the same assumptions that apply to the mutation process modeled by Luria and Delbrück also apply to the conjugation process modeled here. For example, the loss of recipients due to plasmid transfer is ignored in the recipient dynamics of the conjugation model (Eq 6) in the same way that the loss of wild-type cells due to mutation is ignored in the wild-type cell dynamics of the mutation model (Equation 3.1 in Section 3 in S1 File), which tends to be a safe assumption when growth greatly outpaces transformation. Furthermore, in the same way that reversions (mutations restoring a wild-type genotype from a mutant) are disregarded in the mutational model, we ignore the possibility of transconjugants (and donors) becoming plasmid-free through segregational loss in the conjugation model. Lastly, as in the original Luria–Delbrück model, we focus on a pure "birth" process (e.g., once mutants or transconjugants are

generated, their numbers do not decrease). In our supplemental sections, we explore the impacts of violations to some of these assumptions (e.g., the negligible impact of segregational loss in Section 4d in S1 File and how to correct for an effective loss in transconjugant cell numbers due to a failure of small numbers of transconjugants to establish under experimental conditions in Sections 6d and 7 in S1 File). Given the connections between modeling frameworks and estimate structures, we label the expression in Eq 11 as the LDM estimate for donor conjugation rate, where LDM stands for "Luria–Delbrück Method."

## The Luria–Delbrück method (LDM) has improved accuracy and precision

To explore the accuracy and precision of the LDM estimate and compare it to the SIM estimate (as well as other estimates, see Section 4 in S1 File), we used the Gillespie algorithm to stochastically simulate the dynamics of donors, recipients, and transconjugants in a standard mating assay using Eqs 1–3 (Fig 3). Since the mating assay starts without transconjugants, a critical time point (hereafter $t^*$) is marked by the creation of the first transconjugant cell due to the first conjugation event between a donor and a recipient. Before $t^*$, the only events occurring are the cell divisions of donors and recipients (Fig 3A). After $t^*$, all the event types described in Fig 1A can occur. Given that our simulation framework incorporates the stochastic nature of conjugation, $t^*$ will vary among simulated mating assays. One stochastic run of the mating assay constitutes a simulation of the SIM approach. In the laboratory, the standard time point ($\tilde{t}$) used for the SIM estimate is 24 hours; however, a truncated assay ($\tilde{t} < 24$) also produces a nonzero estimate of the conjugation rate as long as the incubation time is greater than $t^*$ (the orange region of Fig 3B and 3C), and the density of transconjugants is large enough to be detected.

While the SIM estimate uses the density of transconjugants ($T_{\tilde{t}}$), the LDM equation instead involves $p_0(\tilde{t})$, the probability that a population has no transconjugants at the end of the assay. A maximum likelihood estimate for this probability (hereafter $\hat{p}_0(\tilde{t})$) is obtained by calculating the fraction of populations (e.g., 100 parallel simulations were used) that have no transconjugants at the specific incubation time $\tilde{t}$ (top of Fig 3D). Thus, the range of time points to calculate the maximum likelihood estimate ($0 < \hat{p}_0(\tilde{t}) < 1$) will be flanking the average $t^*$ (the brown region of Fig 3D). Because the LDM estimate requires the absence of transconjugants in a fraction of populations, while the SIM estimate requires their presence, the range of incubation times for the LDM approach will be earlier than the SIM approach.

Even though there is a range of "valid" incubation times, the accuracy of the SIM estimate can change over time as shown in Fig 3C (same case shown in Fig 1B). In this case, a key modeling assumption of the SIM approach was violated as the transconjugant conjugation rate was much higher than the donor conjugation rate ($\gamma_T \gg \gamma_D$). Consequently, the SIM estimate of the donor conjugation rate was inflated compared to the true value by increasing amounts over time (Fig 3C). In contrast, the LDM estimate under the same scenario had high accuracy and precision over time (Fig 3E). We explored other parameter settings across various incubation times and the LDM estimate generally performed as well or better than other estimates (Section 4 in S1 File).

To more systematically explore the effects of heterogeneous growth and conjugation rates on the accuracy and precision of estimating the donor conjugation rate ($\gamma_D$), we ran sets of simulations sweeping through values of other parameters ($\psi_D$, $\psi_R$, $\psi_T$, and $\gamma_T$). An illustrative example of heterogeneous growth occurs when plasmids confer costs or benefits on the fitness of their host. We simulated a range of growth-rate effects on plasmid-containing hosts from large plasmid costs ($\psi_D = \psi_T \ll \psi_R$) to large plasmid benefits ($\psi_D = \psi_T \gg \psi_R$). Relative to the SIM estimate, the LDM estimate had equivalent or higher accuracy and precision across all

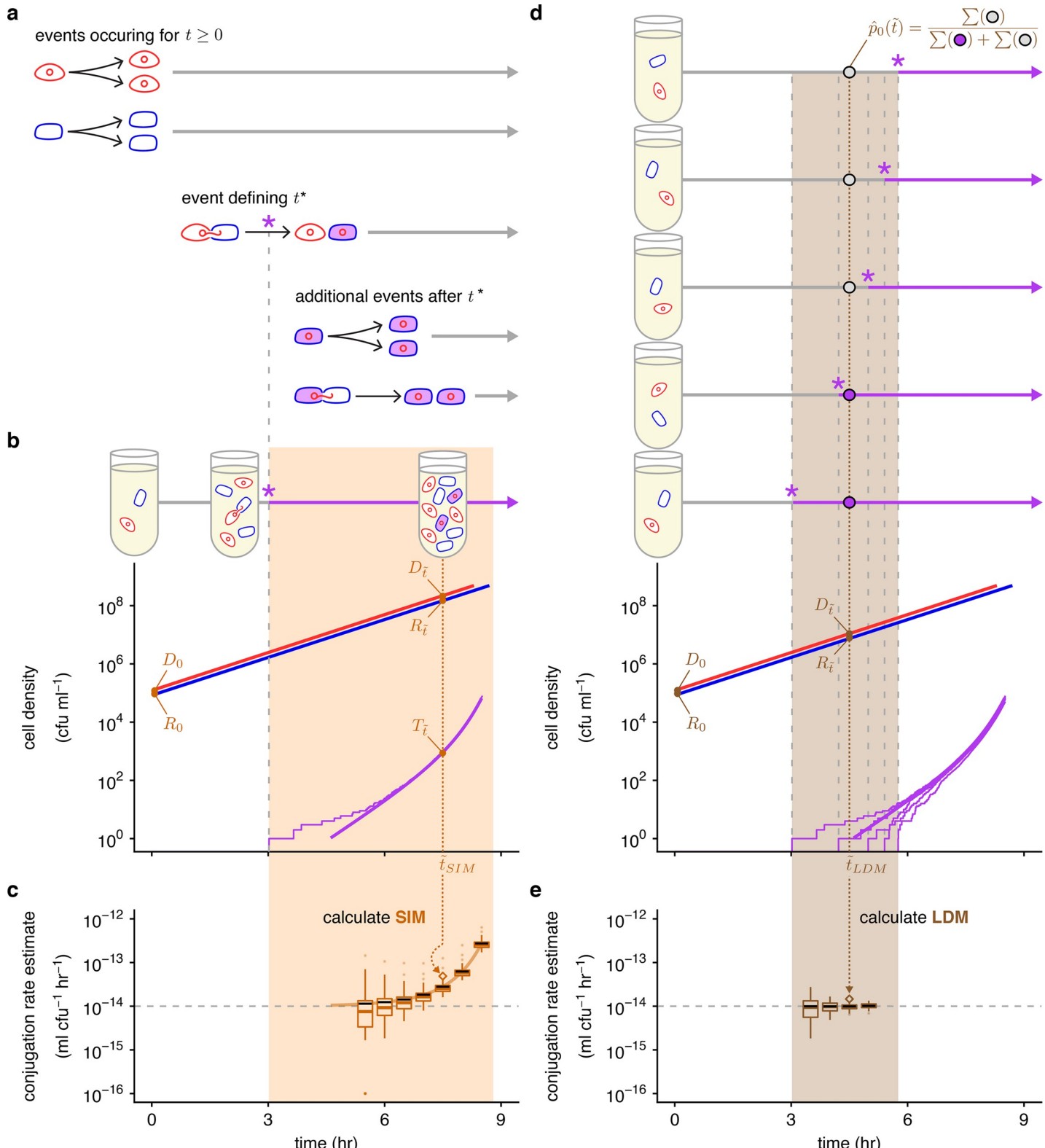

**Fig 3. Overview of stochastic simulation framework and the effects of incubation time on estimating conjugation rate.** (a) The mating assay starts ($t = 0$) with donors and recipients and their populations increase over time. At a critical time ($t^*$, marked by a purple asterisk), the first transconjugant cell is generated through a conjugation event between a donor and recipient. After $t^*$, all possible growth and conjugation events can occur (including transconjugant division and conjugation). (b) A stochastic simulation of Eqs 1–3 shows the donor, recipient, and transconjugant densities (red, blue, and purple thin trajectories, respectively) increasing over

time. The deterministic numerical solution of the same equations and parameter settings from Fig 1B is shown for reference (thick lines). We note that for large densities, the stochastic and deterministic trajectories are closely aligned (i.e., the thick red and blue lines are overlaying their thin counterparts). After a specified incubation time ($\tilde{t}_{SIM}$, dotted orange line), we measure the densities of the 3 populations (orange $D_{\tilde{t}}$, $R_{\tilde{t}}$, and $T_{\tilde{t}}$), which can be used to calculate the (c) SIM estimate. (d) Multiple mating assays are needed for the LDM estimate. Here, 5 stochastic simulations are shown, which display variation in $t^*$. At a specified incubation time ($\tilde{t}_{LDM}$, dotted brown line), we determine the number of assay cultures with transconjugants (purple circles, where for a relevant culture $t^*_i < \tilde{t}_{LDM}$) and without (gray circles, where for a relevant culture, $t^*_j > \tilde{t}_{LDM}$). These numbers are used to calculate $\hat{p}_0(\tilde{t})$, which, along with the donor and recipient densities (brown $D_0$, $R_0$, $D_{\tilde{t}}$, and $R_{\tilde{t}}$), are used for the (e) LDM estimate. The SIM (part c) and LDM (part e) estimates are calculated after different incubation times, where the $\tilde{t}_{SIM}$ (part b) and $\tilde{t}_{LDM}$ (part d) are indicated with orange and brown dotted arrows, respectively. The simulated trajectories in parts b and d would correspond to a single SIM or LDM estimate (the diamond points where the arrows terminate). The light orange and brown backgrounds indicate the range of incubation times giving a finite nonzero estimate of donor conjugation rate for the stochastic runs illustrated in parts b and d. In parts c and e, each box represents the distribution from 100 estimates of the donor conjugation rate for a given $\tilde{t}$, spanning from the 25th to 75th percentile. Given the log y-axis, the zero estimates are placed at the bottom of the y-axis range. The whiskers (i.e., vertical lines connected to the box) contain 1.5 times the interquartile range with the caveat that the whiskers were always constrained to the range of the data. The colored line in the box indicates the median. The solid black line indicates the mean. Parameter values are identical to Fig 1B and used throughout. The data and code needed to generate this figure can be found at https://github.com/livkosterlitz/LDM or https://doi.org/10.5281/zenodo.6677158.

parameter settings (Fig 4A). To explore inequalities in conjugation rate more comprehensively, we simulated a range of transconjugant conjugation rates from relatively low ($\gamma_T \ll \gamma_D$) to high ($\gamma_T \gg \gamma_D$) values. Once again, the LDM estimate generally outperformed the SIM estimate across this range (Fig 4B). In Section 4 in S1 File, we explore other parametric combinations along with model extensions, where, overall, the LDM performed better than the SIM approach and other estimates. Given the large number of simulations for these sweeps, we chose parameter values (i.e., $D_0$, $R_0$, and $\gamma_D$) outside of typical values reported in the literature to reduce the computational burden of the Gillespie algorithm, which occurs when populations sizes become very large. However, the qualitative results were confirmed with a few simulations using parameter settings with more realistic values (Section 4e in S1 File).

## New laboratory protocol to implement the LDM

We developed a general experimental procedure for estimating donor conjugation rate ($\gamma_D$) using the LDM approach in the laboratory. The LDM protocol is tractable and can accommodate a wide variety of microbial species and conjugative plasmids by allowing for distinct growth and conjugation rates among donors, recipients, and transconjugants. The basic approach is to inoculate many donor-recipient cocultures and then, at a time close to the average $t^*$, add transconjugant-selecting medium (counterselection for donors and recipients) to determine the presence or absence of transconjugant cells in each coculture.

In Section 1 in S1 File, we rearrange Eq 11 to provide an alternative form to highlight the quantities needed to conduct the LDM assay in the laboratory:

$$\gamma_D = \frac{f}{\tilde{t}} \left[ -\ln \hat{p}_0(\tilde{t}) \right] \frac{\ln D_{\tilde{t}} R_{\tilde{t}} - \ln D_0 R_0}{D_{\tilde{t}} R_{\tilde{t}} - D_0 R_0}. \tag{13}$$

Similar to previous conjugation estimates, the LDM protocol requires measurement of initial and final densities of donors and recipients ($D_0$, $R_0$, $D_{\tilde{t}}$, and $R_{\tilde{t}}$). In addition, the LDM approach requires a fraction of parallel donor-recipient cocultures to have no transconjugants at the specified incubation time ($\tilde{t}$), which is the maximum likelihood estimate $\hat{p}_0(\tilde{t})$. Lastly, there is a correction factor when the coculture volume deviates from 1 ml; specifically, $f$ is the reciprocal of the coculture volume in milliliters (e.g., for a coculture volume of 100 μl, $f = 1/0.1 = 10$, Section 5 in S1 File).

Before executing the LDM conjugation assay, an incubation time $\tilde{t}$ and initial density for the donors ($D_0$) and the recipients ($R_0$) need to be chosen so that the probability that transconjugants form ($1 - p_0(\tilde{t})$) is not close to 0 or 1. We developed a short assay (Section 6 in S1 File)

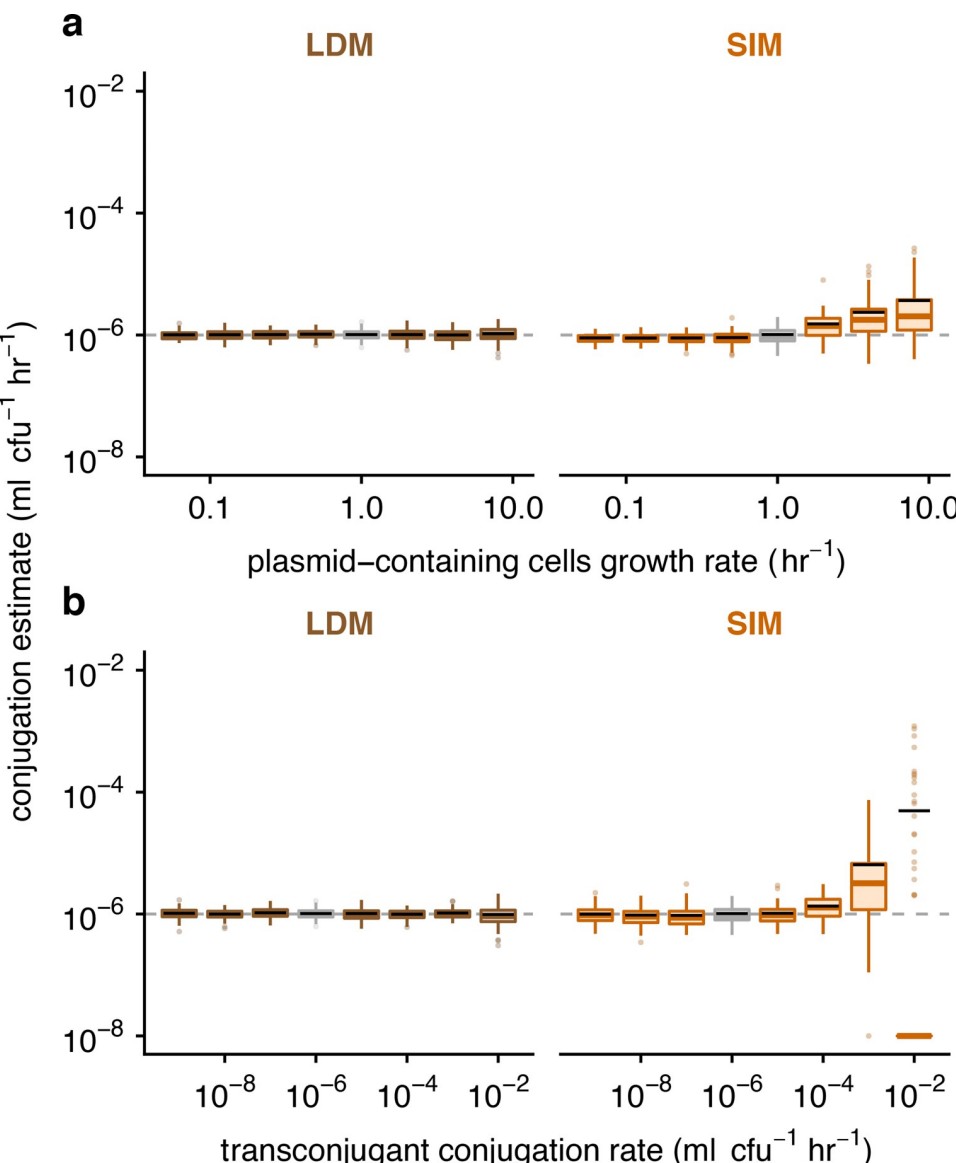

**Fig 4. The effect of parametric heterogeneity on estimating conjugation rate.** The Gillespie algorithm was used to simulate population dynamics. A total of 100 estimates of the donor conjugation rate are shown for each parameter (summarized using boxplots with the same graphical convention as in Fig 3). The gray dashed line indicates the true value for the donor conjugation rate (here, $10^{-6}$). The boxes in gray indicate the baseline parameter setting, and all colored boxes represent deviation of 1 or 2 parameters from baseline. The baseline parameter values were $\psi_D = \psi_R = \psi_T = 1$ and $\gamma_D = \gamma_T = 10^{-6}$. The dynamic variables were initialized with $D_0 = R_0 = 10^2$ and $T_0 = 0$. All incubation times are short but are specific to each parameter setting (see Materials and methods and Table E in S1 File). (a) Unequal growth rates were explored over a range of growth rates for the plasmid-bearing strains, namely $\psi_D = \psi_T \in \{0.0625, 0.125, 0.25, 0.5, 1, 2, 4, 8\}$. (b) Unequal conjugation rates were probed over a range of transconjugant conjugation rates, namely $\gamma_T \in \{10^{-9}, 10^{-8}, 10^{-7}, 10^{-6}, 10^{-5}, 10^{-4}, 10^{-3}, 10^{-2}\}$. For the $10^{-2}$ transconjugant conjugation rate, many of the runs resulted in SIM estimates of zero due to zero transconjugants at the specific incubation time; therefore, the median (colored line) and the box are placed at the bottom of the plot (given that the y-axis is on a log scale). The bulk of the data for this x-value is substantially lower than the mean SIM estimate (black line). The data and code needed to generate this figure can be found at https://github.com/livkosterlitz/LDM or https://doi.org/10.5281/zenodo.6677158.

for screening combinations of incubation time and initial densities to select a target incubation time ($\tilde{t}'$) as well as target initial densities ($D'_0$ and $R'_0$) where $0 < \hat{p}_0(\tilde{t}) < 1$. Note we add primes to indicate that these are "targets" to distinguish $D_0$, $R_0$, and $\tilde{t}$ in Eq 13, which will be

gathered in the conjugation protocol itself. In addition, this pre-assay simultaneously verifies that the LDM modeling assumption of constant growth is satisfied. In our case, this pre-assay revealed several time-density combinations that could have been used. A useful pattern to note is that a higher donor conjugation rate will require shorter incubation times and lower initial densities compared to a lower rate.

For the LDM conjugation assay, we mix exponentially growing populations of donors and recipients, inoculate many cocultures at the target initial densities in a 96 deep-well plate and incubate in nonselective growth medium with the specific experimental culture volume ($1/f$ of 1 ml) for the target incubation time (Fig 5). To estimate the initial densities ($D_0$ and $R_0$), 3 cocultures at the start of the assay are diluted and plated on donor-selecting and recipient-selecting agar plates (Fig 5A). After the incubation time ($\tilde{t}$), final densities ($D_{\tilde{t}}$ and $R_{\tilde{t}}$) are also obtained by dilution plating from the same cocultures (Fig 5B). Liquid transconjugant-selecting medium is subsequently added to the remaining cocultures (Fig 5C). After a long incubation in the transconjugant-selecting medium, there should be a mixture of turbid and nonturbid wells. A turbid well results from one or more transconjugant cells being present at time $\tilde{t}$ (when transconjugant-selecting medium was added). Therefore, a nonturbid well indicates the absence of transconjugant cells at $\tilde{t}$, since the first conjugation event had not yet occurred ($\tilde{t} < t^*$, Fig 3), although see Section 6 in S1 File for a caveat. The proportion of nonturbid cultures is $\hat{p}_0(\tilde{t})$ (Fig 5C). Unlike the traditional Luria–Delbrück method, no plating is required to obtain $\hat{p}_0(\tilde{t})$. With the obtained densities ($D_0$, $R_0$, $D_{\tilde{t}}$, and $R_{\tilde{t}}$), the incubation time ($\tilde{t}$), the proportion of transconjugant-free cultures ($\hat{p}_0(\tilde{t})$), and the experimental culture volume correction ($f$), the LDM estimate for donor conjugation rate ($\gamma_D$) can be calculated via Eq 13.

## Cross-species case study

To empirically test the performance of our assay and our modeling predictions, we initiated a cross-species mating assay between a donor, *Klebsiella pneumoniae* (hereafter "K") with a conjugative IncF plasmid (hereafter "pF"), and a plasmid-free recipient, *Escherichia coli* (hereafter "E"). We denote the donor strain as K(pF), where the host species name is listed first and the plasmid inside the host is given in the parenthesis. E(Ø) denotes the plasmid-free recipient strain. We implemented the LDM and SIM protocols on the same bacterial cultures to compare the laboratory estimates of the cross-species conjugation rate.

The standard SIM protocol involves an incubation of 24 hours. For many bacterial species (including the ones explored here), an incubation time ($\tilde{t}$) of 24 hours will lead to a violation of the assumption of constant growth rates from Eqs 1–3. However, the original study of Simonsen and colleagues did not actually assume constant growth rates [11]. Their model permitted growth rate to vary as a function of resources, but additionally assumed that conjugation rate similarly varied. In other words, the ratio of growth and conjugation rates was assumed to remain constant (Section 1 in S1 File). Under batch culture conditions, the population growth rates will drop as limiting resources are fully consumed (resulting in a stationary phase). If conjugation rates decrease with resources in a similar fashion and the parametric identicality assumptions hold, the SIM estimate can be used over a full-day incubation. Despite these restrictive underlying assumptions, it is not uncommon for researchers to estimate plasmid transfer rates using the 24-hour SIM protocol without experimental validation of assumptions. We proceeded with the standard SIM protocol to allow a comparison between this popular estimate and our new estimate (resulting from the LDM protocol, which does not rest on the same assumptions).

Our LDM estimate of the cross-species conjugation rate was significantly lower than the standard SIM estimate by approximately 3 orders of magnitude (comparison A in Fig 6; *t* test,

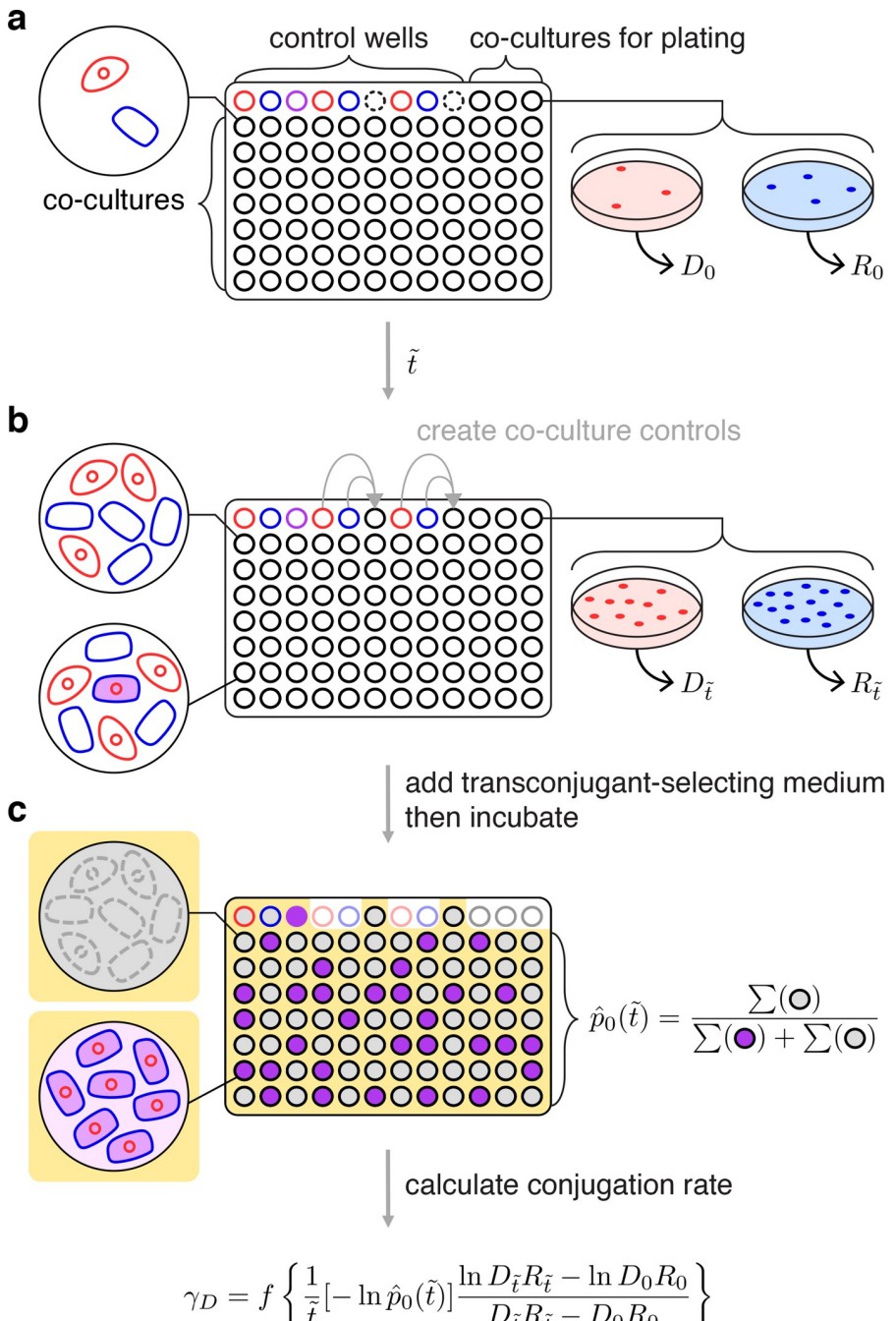

$$\gamma_D = f\left\{\frac{1}{\tilde{t}}[-\ln \hat{p}_0(\tilde{t})]\frac{\ln D_{\tilde{t}}R_{\tilde{t}} - \ln D_0 R_0}{D_{\tilde{t}}R_{\tilde{t}} - D_0 R_0}\right\}$$

**Fig 5. Overview for executing the LDM conjugation protocol.** (a) The wells of a microtiter plate are inoculated with parallel cocultures (black-bordered circles) at the target initial densities ($D'_0$ and $R'_0$). In addition, donor, recipient, and transconjugant monocultures serve as controls (red-, blue-, and purple-bordered wells, respectively). Three cocultures (top-right) are sampled to determine the actual initial densities ($D_0$ and $R_0$). Note empty wells (dash-bordered circles) are used later in the assay. (b) After the incubation time ($\tilde{t}$), the same 3 cocultures are sampled for final densities ($D_{\tilde{t}}$ and $R_{\tilde{t}}$). In addition, donor and recipient monocultures are mixed into the empty wells (indicated by gray arrows) to create coculture controls to verify that diluting with transconjugant-selecting medium effectively prevents conjugation. (c) Subsequently, transconjugant-selecting medium is added to the microtiter plate (indicated by the yellow background) and incubated for a long period. The transconjugant-selecting medium should inhibit donor and recipient growth, leading to nonturbid (gray-filled) donor and recipient control wells, but a turbid (purple-filled) transconjugant control well. In addition, the transconjugant-selecting medium should prevent new conjugation events leading to nonturbid coculture controls (gray-filled). Focusing on the wells inoculated with parallel cocultures, the proportion of transconjugant-free (i.e., nonturbid, gray-filled) cultures is $\hat{p}_0(\tilde{t})$. Using this proportion, the actual

incubation time ($\tilde{t}$), initial densities ($D_0$ and $R_0$), final densities ($D_i$ and $R_i$), and the experimental culture volume correction ($f$), the LDM estimate of the donor conjugation rate ($\gamma_D$) can be calculated. One microtiter plate yields 1 LDM estimate.

$p < 0.0001$). This substantial incongruence could be due to a few possible factors. First, it is possible that the growth and conjugation rates do not change with nutrients in a functionally similar way. While we cannot rule out this possibility, it has been shown for IncF plasmids that both growth and conjugation drop as resources decline to low levels [10], consistent with SIM model assumptions. Second, our growth rate assays (conducted separately from the transfer estimate protocols; see Section 6b in S1 File) revealed our cell types have different growth rates (Fig H in S1 File), thus violating the SIM assumptions. While simulations show there is an effect of these inequalities, the effect size is insufficient to explain the observed difference in comparison A (Fig F in S1 File). Lastly, it is possible that the within-species conjugation, between the E (pF) transconjugants and E(Ø) recipients, occurs at a substantially higher rate than the cross-species conjugation, between the K(pF) donors and E(Ø) recipients. Our simulations show that this kind of difference in conjugation rates can lead to notable inflation of the SIM estimate, and there is evidence that within-species conjugation rates can be markedly elevated over cross-species rates [15,22]. Thus, this last possibility warranted further investigation.

Next, we performed the within-species mating between *E. coli* strains. The LDM estimate for within-species conjugation rate (within *E. coli*) was higher than the cross-species LDM estimate by almost 6 orders of magnitude (comparison B in Fig 6; $t$ test, $p < 0.0001$), a difference that could explain the inflated SIM estimate. To further explore this explanation, we performed an additional cross-species SIM experiment with a shorter incubation time. In Fig 3C, as the incubation time was shortened, the SIM estimate approached the LDM estimate of the donor conjugation rate. Running the SIM protocol with a truncated incubation period (5 hours) resulted in a significantly lower cross-species conjugation rate estimate relative to the standard SIM estimate (comparison C in Fig 6; $t$ test, $p < 0.05$), a result consistent with the pattern predicted under heterogeneous conjugation rates. Under a scenario with reduced parametric heterogeneity (e.g., $\gamma_D \approx \gamma_T$), we predicted that the SIM and LDM estimate would be similar. Consistent with our prediction, the SIM estimate for the within-species conjugation rate using the truncated SIM protocol was not significantly different than the LDM estimate (comparison D in Fig 6; $t$ test, $p = 0.23$).

## Discussion

Conjugation is one of the primary modes of horizontal gene transfer in bacteria, facilitating the movement of genetic material between nonrelated neighboring cells. In microbial communities, conjugation can lead to the dissemination of genes among distantly related species. Since these genes are often of adaptive significance (e.g., antibiotic resistance), a comprehensive understanding of microbial evolution requires a full account of the process of conjugation. One of the most fundamental aspects of this process is the rate at which it occurs. Here, we have presented a new method for estimating the rate of plasmid conjugative transfer from a donor cell to a recipient cell. We derived our LDM estimate using a mathematical approach that captures the stochastic process of conjugation, which was inspired by the method Luria and Delbrück applied to the process of mutation [19]. We explore the connection between mutation and conjugation further in Section 7 in S1 File. Our new method departs from the mathematical approach for other conjugation rate estimates, which assume underlying deterministic frameworks guiding the dynamics of transconjugants [10,11,14]. Beyond the

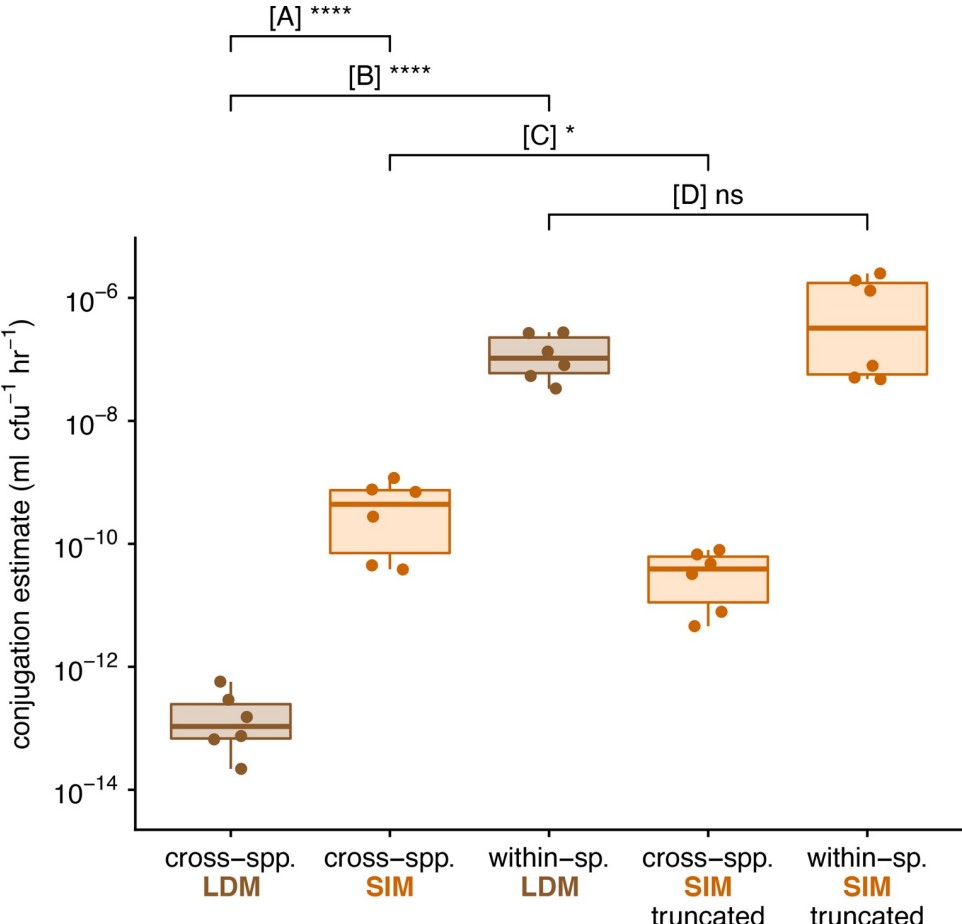

**Fig 6. Experimental estimates for cross-species and within-species conjugation rates.** Each box summarizes 6 replicate estimates by the LDM, SIM, or truncated SIM approach, where each data point corresponds to a replicate. We note each of these estimates involved a correction (see Materials and methods), but the same patterns hold for uncorrected values. (A) Compares the LDM and standard SIM approach for a cross-species mating (between *K. pneumoniae* and *E. coli*). (B) Compares the cross- and within-species mating using the LDM approach. (C) Compares the standard and truncated SIM approach for a cross-species mating. (D) Compares the LDM and truncated SIM approach for a within-species mating. The asterisks indicate statistical significance by a *t* test (1 and 4 asterisks convey *p*-values in the following ranges: $0.01 < p < 0.05$ and $p < 0.0001$, respectively). The data and code needed to generate this figure can be found at https://github.com/livkosterlitz/LDM or https://doi.org/10.5281/zenodo.6677158.

incorporation of stochasticity, the model and derivation behind the LDM estimate relaxes assumptions that constrain former approaches, which makes calculating conjugation rates available to a wide range of systems with different plasmid-donor-recipient combinations.

## The LDM approach has improved accuracy

The most widely used approaches to estimate conjugation rate are derived from the model of Levin and colleagues [10] (discussed in Section 1 in S1 File) that assumes that all strains grow and conjugate at the same rate ($\psi_D = \psi_R = \psi_T$ and $\gamma_D = \gamma_T$). For example, the model underlying the SIM approach assumes precisely this kind of homogeneity. These assumptions and constraints are problematic because bacterial growth and conjugation can and do vary [23,24]. Specifically, donors and recipients are often different taxa and contain chromosomal differences that translate to growth or conjugation rate differences ($\psi_D \neq \psi_R$ or $\gamma_D \neq \gamma_T$). Additionally, plasmid carriage can change growth rate substantially [25], and therefore, recipients can

grow differently from donors ($\psi_R \neq \psi_D$) or transconjugants ($\psi_R \neq \psi_T$). In microbial communities, heterogeneous rates of growth and conjugation are the rule and not the exception. Therefore, a general estimation approach should be robust to this heterogeneity. While the estimates of popular approaches are insensitive to certain forms of heterogeneity, they can also be inaccurate under other forms. In contrast, the LDM estimate remains accurate across a broad range of heterogeneities.

A recent approach by Huisman and colleagues [14] relaxed the assumption of parametric homogeneity, yielding useful revisions to the SIM approach. However, when transconjugants exhibit much larger rates of plasmid transfer than the donors ($\gamma_T \gg \gamma_D$), this new method can become inapplicable. Unfortunately, this kind of difference in conjugation rates is likely not uncommon in microbial communities [15,26]. Indeed, a mating assay involving 2 species can be thought of as a miniaturized microbial community where cross-species conjugation (between donors and recipients) and within-species conjugation (between transconjugants and recipients) both occur. Both previous work [15,26] and experimental data from this study (Fig 6) demonstrate that the transconjugant (within-species) conjugation rate can be significantly higher than the donor (cross-species) rate. In addition, a similar difference in conjugation rates can arise from transitory derepression, a molecular mechanism that temporarily elevates the conjugation rate of a newly formed transconjugant [10,27]. The LDM approach is robust to these differences because it focuses on the creation of the first transconjugant (an event that must be between a donor and recipient) and ignores subsequent transconjugant dynamics (which is affected by transconjugant transfer). The LDM method produces an accurate estimate for donor conjugation rate in systems with unequal conjugation rates, whether the differences are taxonomic or molecular in origin.

## The LDM approach has improved precision

In addition to improved accuracy, the LDM estimate has advantages in terms of precision. Since conjugation is a stochastic process, the number of transconjugants at any given time is a random variable with a certain distribution. Therefore, estimates that rely on the number of transconjugants (which includes nearly all available methods) or the probability of their absence (the LDM approach) will also fall into distributions. Even in cases where the mean (first moment of the distribution) is close to the actual conjugation rate, the variance (second central moment) may differ among estimates. For the number of parallel cocultures in our protocol, the LDM estimate had smaller variance compared to other estimates, even under parameter settings where different estimates shared similar accuracy (e.g., Fig 4). This greater precision likely originates from the difference in the distribution of the number of transconjugants ($T_{\tilde{t}}$) and the distribution of the probability of transconjugant absence ($p_0(\tilde{t})$, where variance decreases with the number of cocultures), something we explore analytically in Section 8 in S1 File. Beyond the mean and the variance, other features of these distributions (i.e., higher moments) may also be important. For certain parameter settings, the estimates relying on transconjugant numbers were asymmetric (the third moment was nonzero). In such cases, a small number of replicate estimates could lead to bias (Section 4 in S1 File). Typically, a small number of conjugation assays is standard; thus, the general position and shape of these estimate distributions may matter. Over the portion of parameter space that we explored, the LDM distribution facilitated accurate and precise estimates through its position (a mean reflecting the true value) and its shape (a small variance and a low skew).

## The LDM approach has implementation advantages

As discussed above, violations of modeling assumptions can lead to significant bias when estimating conjugation rate. Therefore, when implementing a conjugation protocol, the degree to

which the experimental system satisfies the relevant assumptions is of prime importance. The most straightforward way to deal with this issue is to experimentally confirm that assumptions hold. For example, the model underlying the LDM estimate assumes that growth rates of each cell type remain constant throughout the assay. This verification is part of the LDM protocol (see Materials and methods and Section 6 in S1 File). We emphasize that confirming the satisfaction of an assumption for one experimental system does not guarantee that the assumption holds for other systems. For example, the model underlying the SIM estimate assumes that growth and conjugation rates respond in a functionally similar way to changes in resources. While this assumption was explored for the IncF plasmid used in the original SIM study [10], other plasmid systems will readily violate it (e.g., some IncP plasmids conjugate during stationary phase after growth has stopped [28]), which can lead to bias in the estimate (Section 4h in S1 File). Some approaches do not experimentally verify modeling assumptions as part of their corresponding protocol, but rather rely on simulated sensitivity analyses showing violations have little to no effect on the estimate [11,14]. For example, the SIM estimate is robust to relatively small differences in growth rates or conjugation rates [11]. Overall, for any conjugation rate estimate, either the underlying assumptions should be validated for the focal experimental system or a rationale offered for why certain violations by the focal system will not significantly bias the estimate.

Given recent interest in quantitative estimates of conjugation rate [29] and the impacts of model assumption violations [14], there has been a matching interest in altering conjugation protocols such that bias is minimized when violations apply. A common procedural adjustment involves shortening the incubation period of the mating assay because the bias resulting from modeling violations can increase over time [12,13]. For example, when the transconjugant transfer rate is much higher than the donor rate, shortening the incubation time can mitigate some of the inaccuracies in the SIM estimate (Figs 3 and 6). However, there are a few caveats to this adjustment for estimates that rely on transconjugant density (which includes all common approaches, but not the LDM). First, as the incubation time decreases, the benefits in estimate accuracy come at the expense of costs in estimate precision. Specifically, variation in the timing of the first transconjugant cell appearance ($t^*$ in Fig 3) has a greater impact on estimate variance with earlier incubation times. In part because the LDM approach does not rely on a measurement of transconjugant density, the LDM estimate remains both accurate and precise across various incubation times. Second, as incubation time decreases, the transconjugant population can become extremely small, and therefore, technical problems with measuring an accurate transconjugant density through plating can arise [13]. For example, when the transconjugants are rare in the mating culture, the low dilution factor required for selective agar plating for transconjugants ensures a very high density of donors and recipients are simultaneously plated. Before complete inhibition of the donors and recipients by the transconjugant-selecting medium, conjugation events on the plate can generate additional transconjugants inflating the conjugation rate estimate [7,13,30,31]. Recently, a spectrophotometric technique was introduced to avoid selective plating altogether, which addresses this second caveat [13], but not the first. Notably, neither of these 2 caveats applies to the LDM approach because a binary output (turbid or nonturbid cultures) is used in lieu of measuring transconjugant density. Overall, the LDM protocol is both experimentally streamlined and insensitive to factors that can confound other approaches.

## The LDM approach is broadly applicable

In this paper, we have highlighted the possibility that the rate of conjugation may change (substantially) with the identity of the plasmid-bearing cell [32–34]. For example, as a plasmid

moves from the original donor strain to the recipient background (forming a transconjugant), the transfer rate can change (i.e., $\gamma_D \neq \gamma_T$). However, the conjugation rate changes with much more than just the identity of the cell holding the plasmid. The rate of transfer can additionally depend on the identity of the recipient as well as environmental conditions (e.g., level of nutrients, presence of antibiotics) [35]. Thus, there is no single conjugation rate "belonging to" a plasmid-bearing strain. Our LDM approach is meant to be a conditional "snapshot," where the conjugation rate depends on conditions of the protocol and the strains used. It is entirely possible to run the LDM approach under different conditions (e.g., changing nutrients) and assess the effect of environmental factors on transfer rate. The donor conjugation rate can be calculated under any condition as long as strain growth rates are constant over the protocol. But the distinguishing feature that gives the LDM method relative breadth of application is that it is robust to a form of conditionality that is tied to the mating assay itself. Specifically, because transconjugants are formed during a mating assay and, like donors, can deliver the plasmid to additional recipients, a form of rate conditionality is an unavoidable possibility for any protocol employing a mating assay. As we have shown (Figs 1, 3, 4 and 6), a difference in transfer rate between donors and transconjugants can make popular estimates inaccurate. However, by focusing on the first transconjugant formed (which only involves the donor and recipient, Fig 6), the LDM sidesteps this conditionality altogether, allowing an unbiased estimate of donor conjugation rate under a user-defined environment.

In conclusion, the LDM offers new possibilities for measuring the conjugation rate for many types of plasmids, species, and environmental conditions. We have presented evidence that supports our method being more accurate and precise than other widely used approaches. Importantly, the LDM eliminates bias caused by relatively high transconjugant conjugation rates, which is not unlikely when the donor and recipient belong to different species. We experimentally explored a case where the transconjugant transfer rate was dramatically higher than the donor rate and found that a standard estimate could inflate the conjugation rate (Fig 6). More generally, violations of model assumptions, intrinsic stochasticity, and implementation constraints can cause problems for currently available approaches. However, an adjustment of the approach Luria and Delbrück used to explore and estimate mutation nearly 80 years ago can address many of these issues. This new approach greatly expands the ability of experimentalists to accurately measure conjugation rates under the diverse conditions found in natural microbial communities.

## Materials and methods

More detailed information for the mathematical models, simulations, and experiments is provided in the Supporting information in S1 File. A general LDM protocol is deposited on protocols.io (dx.doi.org/10.17504/protocols.io.e6nvwk812vmk/v3).

### Bacterial strains

Donor strains included 2 Enterobacteriaceae species: *Escherichia coli* K-12 BW25113 [36] and *Klebsiella pneumoniae* Kp08 [7]. We use the first letter of the genus (E and K) to refer to these species throughout. The recipient strain is derived from the same isogenic strain as the *E. coli* donor strain but encodes additional chromosomal streptomycin resistance, providing a unique selectable marker to distinguish the donor and recipient hosts in both the cross- (K to E) and within-species (E to E) mating assays. The focal conjugative plasmid was used previously [37]: plasmid F'42 from the IncF incompatibility group. A tetracycline resistance gene was previously cloned into the F'42 plasmid [38] and used as the selectable marker to distinguish plasmid-containing from plasmid-free hosts. This derived plasmid is referred to as "pF" throughout.

## Conjugation assays

Strains were inoculated into LB medium from frozen isogenic glycerol stocks and grown for approximately 24 hours. The plasmid-containing cultures were supplemented with 15 μg ml$^{-1}$ tetracycline to maintain the plasmid. The saturated cultures were diluted 100-fold into LB medium to initiate another 24 hours of growth (to acclimate the previously frozen strains to laboratory conditions). The acclimated cultures were then diluted 10,000-fold into LB medium and incubated for 4 hours to ensure the cultures entered exponential growth (Section 6b in S1 File). The exponentially growing cultures were diluted by a factor specific to the donor-recipient pair (Section 6e in S1 File), mixed at equal volumes, and dispensed into 84 wells of a deep-well microtiter plate at 100 μl per well (Fig 5a solid black-bordered wells in rows 2 to 8, these wells were the cocultures used to estimate $p_0(\tilde{t})$). In an additional 3 wells (Fig 5a solid black-bordered wells in top row), 130 μl (per well) of the mixture was dispensed, and immediately 30 μl was removed to determine the initial densities ($D_0$ and $R_0$) via selective plating. An additional 3 wells contained monocultures of the 3 strains. Specifically, 100 μl of donor, recipient, and transconjugant cultures were placed in their own well (Fig 5A red-, blue- and purple-bordered wells, respectively, in the top left). Later in the assay, these monocultures determined if the transconjugant-selecting medium prohibited growth of both donors and recipients, while permitting growth of transconjugants. An additional 4 wells contained diluted monocultures of donors and recipients (2 wells each at 100 μl, Fig 5A red- and blue-bordered wells, respectively, in the top middle). These monocultures were used to create cocultures (in empty wells, Fig 5A dash-bordered wells) during the assay itself (see below). The deep-well plate was incubated for a predetermined incubation time $\tilde{t}$ (Section 6e in S1 File), after which 3 events occurred in rapid succession. First, 30 μl was removed from each of the 3 wells used to determine initial densities to uncover the final densities ($D_{\tilde{t}}$ and $R_{\tilde{t}}$) via selective plating (Fig 5B). We note that densities were calculated from each well then averaged for calculating the LDM estimate. Second, donor and recipient monocultures were mixed at equal volumes into the 2 empty wells (Fig 5B, gray arrows). At a later point in the assay, these 2 wells verified that new transconjugants did not form via conjugation after transconjugant-selecting medium was added. Third, 900 μl of transconjugant-selecting medium (7.5 μg ml$^{-1}$ tetracycline and 25 μg ml$^{-1}$ streptomycin; see Section 6c and 6d in S1 File) was added to all cocultures used to estimate $p_0(\tilde{t})$ as well as relevant control wells (Fig 5C, yellow background). This medium disrupted new conjugation events—immediately by diluting cells then by inhibiting donors and recipients—while simultaneously allowing growth of transconjugants. When designing transconjugant-selecting media, appropriate preliminary and control experiments must be conducted to ensure that the media enables exclusive growth of transconjugants (see Section 6c in S1 File). The deep-well plate was incubated for 4 days, and the state of all wells (turbid or nonturbid) was recorded. For both mating assays in this study (i.e., cross- and within-species), this conjugation protocol was repeated 6 times.

For the cross-species mating, the SIM method was executed alongside the LDM method described above. The SIM approach was conducted for 2 incubation times: a standard 24 hours and a truncated 5 hours. In an additional deep-well plate, 100 μl of the donor-recipient coculture was dispersed into 6 wells, split into 2 groups of 3 wells where each group corresponded to a different incubation time. To derive the SIM estimate for each incubation group, 30 μl was removed from each of the 3 wells in the group at the relevant incubation time ($\tilde{t} = 5$ and $\tilde{t} = 24$) to determine the final donor ($D_{\tilde{t}}$), recipient ($R_{\tilde{t}}$), and transconjugant ($T_{\tilde{t}}$) densities via selective plating. Densities were calculated from each well then averaged for calculating the SIM estimate. This protocol was repeated 6 times alongside the LDM replicates. For the within-species mating, the SIM method was executed as outlined above for the cross-species

mating but using only the truncated SIM method with a 3-hour incubation time. Similar to the LDM protocol, we ran a control to confirm that conjugation did not occur after cocultures were exposed to transconjugant-selecting medium, but in this case, it was for agar plates instead of liquid medium. Specifically, for the first SIM replicate per incubation time, an additional 3 donor monocultures and 3 recipient monocultures were initiated as above. At each incubation time (e.g., 5 and 24 hours for the cross-species mating), 3 new donor-recipient cocultures were created in empty wells and immediately plated on transconjugant-selecting agar at dilutions used to determine transconjugant densities. For this case, no transconjugant colonies formed (indicating that conjugation does not occur on the selective agar plate). We emphasize that this is a necessary step for any new system as post-plating conjugation has been reported [7,13,31].

For both the LDM and SIM approaches, the working assumption is that a cell will successfully establish a lineage under the appropriate selective conditions. As one example, a well with a single transconjugant will become turbid after incubation with transconjugant-selective medium. As another example, a donor cell on a donor-selecting agar plate will form a visible colony after incubation. A recent paper [39] has clearly demonstrated that this working assumption needs to be checked. In Section 6 in S1 File, we offer adjustments to the protocols to improve the chances that this assumption holds. Additionally, we present ways to correct estimates if the assumption does not hold. In Fig 6, we used these corrections (see Sections 6 and 7 in S1 File for details).

## Stochastics simulations

We used the Gillespie algorithm available in the GillesPy2 open-source Python package for stochastic simulations [40]. We specified starting cell densities and parameters and simulated population dynamics using Eqs 1–3 for a set incubation time in a 1-ml culture volume. For each parameter setting, we simulated 10,000 populations and calculated the conjugation rate using the LDM and SIM estimates. Each estimate has different requirements for calculating the conjugation rate (Fig 3). The LDM estimate needs multiple populations to calculate $\hat{p}_0(\tilde{t})$; therefore, for each LDM estimate, we reserved 100 independent populations to compute $\hat{p}_0(\tilde{t})$, then 1 random population in the set of 100 was used to calculate the initial and final cell densities. In other words, the 10,000 populations yielded 100 LDM estimates. In contrast, 1 simulated population yields 1 SIM estimate. Therefore, we used the random populations chosen to calculate the densities for each of the 100 LDM estimates to calculate 100 SIM estimates.

For the incubation time sweeps (Fig 3), the conjugation rate was estimated at 30-minute intervals up until the total population size reached $10^9$ cfu ml$^{-1}$. A 30-minute interval was analyzed if at least 90% of the estimates were finite and nonzero. Notably, the 30-minute intervals occur over an earlier time range for the LDM estimate then for the SIM estimate due to the different estimate requirements.

To compare across various parameter settings (Fig 4), a single incubation time was chosen per parameter set and type of estimate (see Table E in S1 File for the incubation times used). For each parameter setting, the incubation time $\tilde{t}$ for the LDM estimate is set to the average $t^*$. In addition, the incubation time for the SIM estimate is given by the time point for which an average of 50 transconjugants is reached. This choice resulted in a truncated SIM approach (i.e., $\tilde{t} < 24$). However, any estimate bias from a truncated simulation would be conservative relative to the standard SIM approach.

In Section 9, we explore through further simulation the impacts of the random effects of dilution, plating, and lineage extinction on the accuracy and precision of the LDM and SIM approaches.

## Supporting information

**S1 File. Supporting information.**
(PDF)

## Acknowledgments

We thank Hannah Jordt and members of the Kerr and Top laboratories for useful suggestions on the manuscript.

## Author Contributions

**Conceptualization:** Olivia Kosterlitz, Benjamin Kerr.

**Data curation:** Olivia Kosterlitz.

**Formal analysis:** Olivia Kosterlitz, Ivana Bozic, Benjamin Kerr.

**Funding acquisition:** Eva M. Top, Benjamin Kerr.

**Investigation:** Olivia Kosterlitz, Adamaris Muñiz Tirado, Claire Wate, Clint Elg, Benjamin Kerr.

**Methodology:** Olivia Kosterlitz, Benjamin Kerr.

**Project administration:** Olivia Kosterlitz, Eva M. Top, Benjamin Kerr.

**Resources:** Olivia Kosterlitz, Benjamin Kerr.

**Software:** Olivia Kosterlitz, Benjamin Kerr.

**Supervision:** Eva M. Top, Benjamin Kerr.

**Validation:** Olivia Kosterlitz, Adamaris Muñiz Tirado, Claire Wate, Ivana Bozic, Benjamin Kerr.

**Visualization:** Olivia Kosterlitz, Benjamin Kerr.

**Writing – original draft:** Olivia Kosterlitz, Benjamin Kerr.

**Writing – review & editing:** Olivia Kosterlitz, Adamaris Muñiz Tirado, Claire Wate, Clint Elg, Ivana Bozic, Eva M. Top, Benjamin Kerr.

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
