## [Editor Report · Decision Letter 0]

11 Feb 2022

Dear Dr Kosterlitz, 

Thank you for submitting your manuscript entitled "Estimating the rate of plasmid transfer with an adapted Luria-Delbrück fluctuation analysis" for consideration as a Methods and Resources by PLOS Biology.

Your manuscript has now been evaluated by the PLOS Biology editorial staff, as well as by an academic editor with relevant expertise, and I'm writing to let you know that we would like to send your submission out for external peer review.

Once your full submission is complete, your paper will undergo a series of checks in preparation for peer review. Once your manuscript has passed the checks it will be sent out for review. To provide the metadata for your submission, please Login to Editorial Manager (https://www.editorialmanager.com/pbiology) within two working days, i.e. by Feb 15 2022 11:59PM.

If your manuscript has been previously reviewed at another journal, PLOS Biology is willing to work with those reviews in order to avoid re-starting the process. Submission of the previous reviews is entirely optional and our ability to use them effectively will depend on the willingness of the previous journal to confirm the content of the reports and share the reviewer identities. Please note that we reserve the right to invite additional reviewers if we consider that additional/independent reviewers are needed, although we aim to avoid this as far as possible. In our experience, working with previous reviews does save time. 

If you would like to send previous reviewer reports to us, please email me at rroberts@plos.org to let me know, including the name of the previous journal and the manuscript ID the study was given, as well as attaching a point-by-point response to reviewers that details how you have or plan to address the reviewers' concerns. 

Given the disruptions resulting from the ongoing COVID-19 pandemic, please expect some delays in the editorial process. We apologise in advance for any inconvenience caused and will do our best to minimize impact as far as possible.

Kind regards,

Roli Roberts

Roland Roberts

Senior Editor

PLOS Biology

rroberts@plos.org

---

## [Decision Letter · Decision Letter 1]

7 Apr 2022

Dear Dr Kosterlitz,

Thank you for submitting your manuscript "Estimating the transfer rates of bacterial plasmids with an adapted Luria-Delbrück fluctuation analysis" for consideration as a Methods and Resources at PLOS Biology. Your manuscript has been evaluated by the PLOS Biology editors, an Academic Editor with relevant expertise, and by three independent reviewers.

You'll see that all three reviewers are broadly very positive about your study, but each raises a number of issues that should be addressed before further consideration (do note that most of reviewer #3's comments are contained in the attached file). In light of the reviews (below), we are pleased to offer you the opportunity to address the [comments/remaining points] from the reviewers in a revised version that we anticipate should not take you very long. We will then assess your revised manuscript and your response to the reviewers' comments and we may consult the reviewers again.

We expect to receive your revised manuscript within 1 month.

**IMPORTANT - SUBMITTING YOUR REVISION**

*Resubmission Checklist*

*Published Peer Review*

*PLOS Data Policy*

*Blot and Gel Data Policy*

Sincerely,

Roli Roberts

Roland Roberts

Senior Editor

PLOS Biology

rroberts@plos.org

REVIEWERS' COMMENTS:

Reviewer #1:

The authors have developed a new method to estimate plasmid conjugation rates that explicitly accounts for the stochasticity inherent in conjugation. This is a fundamentally new method, and a refreshing take on estimating conjugation rates. I think this will be a useful tool for microbiologists and a good starting point for further work taking into account the stochasticity of conjugation.

I would also like to congratulate the authors on their development of the paper since the very first preprint version. However, I see several points that could further improve this paper and heighten its impact.

-- General comments: -- 

1. The LDM method is unique in its focus on the stochastic, early dynamics of conjugation. I am missing some more discussion of the biases unique to these early measurements. The authors already discuss the transconjugant extinction probability (in the presence of antibiotics), but it is a bit hidden in the supplementary materials. I did not see any discussion of dilution or plating errors (which may be greater at smaller population sizes), nor population level heterogeneity in resumption of growth when starting the conjugation assay. How do these stochastic dynamics affect the LDM in a way that is similar/different to other methods?

2. In the stochastic simulations different numbers of simulations are compared for the different methods, since the LDM requires a panel of 84 simulations to be run for a single p0 estimate (line 652-658). This should be made equal to truly compare the variability of the different methods. From the description it was also not directly clear whether p0 was computed independently multiple times or reused across simulations to estimate the LDM. I would suggest comparing 1000 simulated conjugation assays of TDR / SIM / ASM against 1000 x 84 simulations to estimate the LDM.

3. The substantial difference between the SIM and LDM on the experimental data (Fig. 6) is surprising, and I am missing the base case that shows they perform the same on a scenario where they should both do well (i.e. same growth/conjugation rates). The corresponding interpretation of the results (Line 367-382) also feels highly speculative. In the Figure 6 I further find it confusing that the cross-spp. SIM truncated is not listed next to the other cross-spp. results; and that there is no within-sp. SIM result. 

I think this is an important point because the current Fig gives the feeling that "different methods simply give different results", which would hinder the broader adoption of the LDM and the comparison of results across the literature.

-- Specific comments: --

a. The abstract and significance statement do not do the rest of the paper justice. Specifically the significance of this manuscript to me seems more in the stochastic treatment of conjugation than in showing that commonly used methods are biased. Further, the results apply to all plasmids, not just those that carry antibiotic resistance.

b. Fig 3: 

- Panel A: the placement of the label "events occuring after t*" seems to suggest these are the only events occuring after t*. Perhaps the authors could move the label to above all lines/types of events, or reword to "additional events occuring after t*"

- Panel C vs E: related to the general comment above, this seems an "unfair" comparison of both methods since these are not 10'000 independent estimates of the LDM. Line 744: I would perhaps rephrase that these are calculated "After" instead of "for" different incubation periods.

c. Line 241-242: it may help readers to reiterate that this intuition only holds early on during the mutational dynamics, when the R population is not too affected by transformation into transconjugants yet.

d. Line 258-259: this statement is quite simplified, and ignores the fact that the SIM also requires the population sizes to be large enough to be detectable.

e. Line 260-269: At this point in the text it was not clear how many wells/simulations are needed for the maximum likelihood estimate of p0. Some info on this can be gleaned from SI Fig. 10, but it would be helpful to know how many wells are necessary for the LDM estimates to be accurate (like the convergence). Similarly, it would be good to know whether this depends on how close p0(t) is to 0 or 1 (how sensitive is the LDM to the statement in line 316-318).

f. Line 292-294: This description is unclear and would benefit from stating more explicitly that the Gillespie algorithm gets slow when population sizes are large (i.e. that the difference between both parameter sets is in D0, R0 and gammaD). 

g. Cross-species case study (line 366): when and how was the growth rate of the individual strains determined? 

Line 358-365: The authors did not test that the resource dependence is the same for conjugation and growth. As such, this paragraph seems to suggest a generality that doesn't hold, and the SIM estimate may be unnecessarily bad by running the assay over 24 instead of shorter (e.g. 6 hours).

h. Line 494: Which violations of model assumptions is meant for the Dewar et al study here?

i. Line 530-536: I do not understand this section about the functional response and the conditionality of different methods. Could the authors explain their argument in different words?

j. Line 591: The number of 84 wells does not seem to match the number of black bordered wells in Fig. 5 (87 wells).

k. Line 638-646: I would like to see this discussion on establishment probability (+ perhaps stochasticity in pipetting etc.) more prominently discussed in the main text. In particular I would assume the effect of this stochsatic extinction is different for the SIM estimate vs. the LDM, but it would be good to test this with simulations.

l. Line 656: It would be helpful to mention this information on the number of populations used to estimate the LDM (vs. other methods) also in the main text (e.g. around line 275) and/or in the figure captions.

m. Fig. 4: why did the runs result in SIM estimates of zero? Was the recipient population 0 in these simulations? 

n. SI table 3: The text reads "The SIM model can incorporate resource-dependent growth and conjugation *because* (1) growth and transfer rates are homogeneous", but I would rather think this holds *if/when* these two conditions are given.

o. Github folder:

- I could not find the code that was used to produce the simulations that are shown in SI Fig. 1 (or Fig. 1 for that matter), only the code to plot those simulation results.

- Some readme pages contain comments that look like they were meant to be internal/private (e.g. https://github.com/livkosterlitz/LDM/tree/main/Figures ), and others don't display well (https://github.com/livkosterlitz/LDM/tree/main/Simulations and https://github.com/livkosterlitz/LDM/tree/main/Supporting_data).

p. SI Fig 1-5: 

- The TDR estimate is consistently a factor 10 smaller than the other estimates. Can the authors explain why?

- Some legends read "Zero estimates were set to 10^-9": why were these 0? One could argue that in this scenario and experimenter would clearly see something is wrong, and would redo the experiment for a different incubation period / initial pop sizes.

- Do I understand the Material and Methods correctly that the incubation period was changed for each method depending on when a certain # tranconjugants were reached, i.e. dependent on the transconjugant growth rate + the conjugation rate (e.g. separately for each x-position in Fig S1)?

- The critical time threshold of the ASM is mentioned only for SI Fig. 3 (Line 386-390) but it looks like this may have been a factor already in SI Fig. 2.

q. Line 376: It was insufficiently clear that this reference to SI section 4 refers to sections 4fgh, and within those sections not so clear what the assumptions and outcomes are (+ how they relate to the cross-species conjugation experiment).

- How good is the assumption (SI Line 476/477) that conjugation and plasmid loss are Monod functions of the resource concentration?

- Caption of Fig S6: have these significant deviations (Panel b/c) been tested statistically? Are the parameter estimates from the LDM?

- Is Fig S7 repeated with the same parameters as Fig S6? The descriptive text is written as if this is general behaviour of the SIM, but the depletion of the recipient pool seems parameter-specific?

Reviewer #2:

In this manuscript, Kosterlitz and colleagues propose an elegant and effective method for estimating plasmid conjugation rates. The key innovation, inspire by Luria and Delbrück's classic of (microbial) evolutionary genetics, is to use presence/absence of transconjugants across multiple cultures, rather than absolute counts from a single culture. Simulations and data convincingly show that the proposed method is more robust than existing approaches, and the authors provide extensive (re-)derivations enabling comparison with the state-of-the-art. Overall I thought this was an excellent piece of work and makes a valuable contribution to the field. The analyses are comprehensive, but clearly written and easy to follow even for a non-mathematician. Supplementary Information and Figures are excellent. I do, however, have some suggestions/comments for consideration, which may improve things further.

1. In the proposed LDM approach, p[hat]_0(t[tilde]) is the probability of zero mutants at time t[tilde] and is estimated by establishing multiple cultures and counting the proportion of negatives. p[hat]_0(t[tilde]) is therefore an estimate, the accuracy of which will presumably improve with the number of cultures established. There is not much discussion/analysis of how the accuracy of the overall measurement will vary with the number of cultures. This has consequences for experimental design. While the recommended 96-well plate design is sensible, I wonder if there is a minimum number of cultures that is required for accurate calculation? Or is it worth investing more resources to run more cultures to improve accuracy? Is the proposed one-plate-per-measurement approach the most efficient?

2. Following: in a traditional conjugation assay (e.g. SIM), one would normally establish multiple replicate conjugation reactions for each treatment, enumerate the various (sub-)populations within each replicate, and calculate conjugation rate from each replicate. The mean and variance of these calculated values would then be statistically analysed and compared between conditions e.g. by a linear model. A similar approach appears to have been used for the LDM in Figure 6 to compare between these approaches — 6 replicates were run (lines 616-617). But perhaps, if one was to exclusively use the LDM method to compare treatments, it would be possible to test statistical significance on a basis analogous to a binomial test? That is, rather than running 84 wells six times (n = 6), a measure of variance could be calculated from the proportion of positives across all 504 wells. I'm not sure exactly how this would work in the context of the full formula, nor whether it would increase statistical power... I accept that addressing this may well be beyond the scope of the current manuscript but I am curious to know the authors' thoughts. 

3. The authors comprehensively and clearly address the potential issues in their approach arising from transconjugant-selecting media and extinction probability in the SI (SI section 6d). However this was not well signposted in the main text. Perhaps a couple of words could be inserted line 615: "When designing transconjugant-selecting media, appropriate preliminary and control experiments must be conducted to ensure that the media enables exclusive growth of transconjugants (see SI section 6d)".

Reviewer #3:

[IMPORTANT - SEE ATTACHMENT FOR FULLY FORMATTED REVIEW]

Very impressive manuscript - I have only minor (mostly discretionary) comments. Please see attachment for details.

---

## [Decision Letter · Decision Letter 2]

11 Jun 2022

Dear Dr Kosterlitz,

Thank you for your patience while we considered your revised manuscript "Estimating the transfer rates of bacterial plasmids with an adapted Luria-Delbrück fluctuation analysis" for publication as a Methods and Resources at PLOS Biology. This revised version of your manuscript has been evaluated by the PLOS Biology editors, the Academic Editor, and two of the original reviewers.

Based on the reviews, we're likely to accept this manuscript for publication, provided you satisfactorily address the remaining points raised by the reviewers and the following data and other policy-related requests.

IMPORTANT:

a) Please attend to the remaining requests from reviewer #3.

b) Thanks for providing the underlying data and code in GitHub. Please could you also deposit a permanent, unchangeable version in a repository like Zenodo that provides a DOI?

c) Please cite these (Github/Zenodo etc) clearly in each relevant main and supplementary Figure legend as the location of the data (e.g. "The data and code needed to generate this Figure can be found at https://github.com/livkosterlitz/LDM etc.").

We expect to receive your revised manuscript within two weeks. 

*Published Peer Review History*

*Press*

Sincerely,

Roli Roberts

Roland Roberts, PhD

Senior Editor,

rroberts@plos.org,

PLOS Biology

DATA NOT SHOWN?

REVIEWERS' COMMENTS:

Reviewer #2:

I am satisfied with the author responses to my previous review and am very happy to recommend this manuscript for publication. 

Reviewer #3:

The authors have thoroughly addressed my previous comments. I especially like the additional analysis added to SI section 8 regarding optimising experimental design for a fixed total number of wells.

A few minor issues for final polishing:

- The authors could consider mentioning in the main text that the protocol is detailed on protocols.io

- Regarding my previous comment #13, the authors gave a detailed response with additional analysis, but unless I've overlooked something, it doesn't look like anything was changed in the manuscript. The authors could consider at least adding their final conclusion, that variances of the SIM and ASM estimates are approximately equal. (Perhaps the full derivation could be added to the GitHub Appendix?)

- In the newly added SI Figure 11: 

In caption (a), variation  variance?; "among the 100 estimates" rather than "of each estimate"?

In panel (d), how is \\tilde{t} set?

"Each partitioning was run 10 times" - was a new set of 500 populations simulated for each partitioning? (How else would you get variation among the 10 partitioning when W=500?)

- Regarding my previous comment #29: I'm guessing the authors mean to say a dilution factor of 4 x 10^7, not 10^-7?

---

## [Editor Report · Decision Letter 3]

29 Jun 2022

Dear Dr Kosterlitz,

Thank you for the submission of your revised Methods and Resources paper "Estimating the transfer rates of bacterial plasmids with an adapted Luria-Delbrück fluctuation analysis" for publication in PLOS Biology. On behalf of my colleagues and the Academic Editor, Wenying Shou, I'm pleased to say that we can in principle accept your manuscript for publication, provided you address any remaining formatting and reporting issues. These will be detailed in an email you should receive within 2-3 business days from our colleagues in the journal operations team; no action is required from you until then. Please note that we will not be able to formally accept your manuscript and schedule it for publication until you have completed any requested changes.

Sincerely,

Roli Roberts

Senior Editor

PLOS Biology

rroberts@plos.org